# Improvement of the Computational Efficiency in SVD-3DEnVar Data Assimilation Scheme and Its Preliminary Application to the TRAMS 3.0 Model

Kun Liu<sup>1,2</sup>, Daosheng Xu<sup>3,\*</sup>, Fei Zheng<sup>2</sup>, Juanxiong He<sup>2</sup>, Chun Li<sup>1</sup>, Jeremy Cheuk-Hin Leung<sup>3</sup>, Mingyang Zhang<sup>3</sup>, Dingchi Zhao<sup>3</sup>, Quanjun He<sup>4</sup>, Yuewei Zhang<sup>4</sup>, Yi Li<sup>3</sup>, Banglin Zhang<sup>3</sup>

Correspondence to: Daosheng Xu (dsxu@gd121.cn)

Abstract. Although the Singular Value Decomposition-three Dimensional Ensemble Variational (SVD-3DEnVar) data assimilation scheme has achieved successful application in real case simulations with comprehensive numerical weather prediction models, its computational efficiency still cannot meet the demands of actual operational numerical forecasting. The main limitations lie in the generation of three-dimensional perturbations and the implementation of parallel calculations. This paper constructed a three-dimensional perturbation field generation scheme that supports multi-process parallelism and can directly generate any specified number of grid points in both horizontal and vertical directions. At the same time, an efficient parallel implementation scheme has been developed according to the characteristics of local patch assimilation in the SVD-3DEnVar scheme. The Observing System Simulation Experiment (OSSE) test results based on the Tropical Regional Atmospheric Model System (TRAMS) show that after computational efficiency optimization, the time required to generate a 3D perturbation field has been reduced from 22 minutes to 2.2 seconds, while the runtime of the assimilation process has decreased from 1,700 minutes under serial execution to less than 15 minutes (using 150 nodes in parallel). Finally, we conducted an assimilation experiment using actual observational data of sea surface wind fields to preliminarily validate the reasonableness of the assimilation results from the optimized SVD-3DEnVar scheme.

#### 25 1 Introduction

The accuracy of typhoon numerical forecasting is highly sensitive to the quality of the initial conditions. The accuracy of the large-scale environmental flow surrounding a typhoon largely determines its subsequent track; meanwhile, initial errors in the mesoscale and convective systems within the typhoon core can rapidly amplify and affect the predictability of typhoon intensity (Wang and Wu, 2004; Weng and Zhang, 2012; Xu et al., 2025a). Data assimilation, by effectively integrating observations from multiple platforms such as satellites, radars, and dropwindsondes with the background field (short-range

<sup>&</sup>lt;sup>1</sup>College of Oceanic and Atmospheric Sciences, Ocean University of China, Qingdao, 266100, China
<sup>2</sup>Institute of Atmospheric Physics, Chinese Academy of Sciences, Beijing, 100029, China
<sup>3</sup>College of Atmospheric and Oceanic Sciences, National University of Defense Technology, Changsha, 410003, China

<sup>&</sup>lt;sup>4</sup>Guangzhou Meteorological Satellite Ground Station, Guangzhou, 510650, China

35

forecasts) of numerical models, can produce an initial state that is physically consistent and closer to the true atmosphere, including a realistic typhoon vortex structure and its surrounding environment (e.g., steering flow), thereby enhancing the ability of numerical models to forecast typhoon track and intensity (Bauer et al., 2015; Wu et al., 2007). With the continuous increase in satellite observations in recent years, the importance of assimilating these data to improve typhoon forecasts has become increasingly evident (Li et al., 2019; Xiao et al., 2023; Yang et al., 2018; Zhang et al., 2022b).

Typhoon observations over the ocean primarily rely on satellite data. The nonlinear nature of the observation operators and the high temporal and spatial resolution characteristics of these observations place high demands on the performance of data assimilation schemes. Therefore, the continuous improvement of data assimilation techniques is one of the key factors driving the enhancement of typhoon numerical forecasting accuracy. In early operational typhoon forecasting, 3-Dimensional Variational (3DVAR) scheme was commonly used for data assimilation. The main limitation of this method is that it uses static, climatological background error covariance to propagate observation information (Lorenc, 2003; Bannister, 2008). This poses significant limitations when dealing with strong nonlinear and rapidly evolving weather systems like typhoons. For instance, it may erroneously propagate information from high-level clouds to clear lower-level areas, resulting in spurious structures in the analysis field, which in turn affects the improvement of typhoon forecasts through data assimilation. To overcome the limitation of 3DVar, which cannot account for the flow-dependent nature of background error covariance, several improvements were proposed, including four-dimensional variational assimilation and ensemble-variational hybrid assimilation methods, leading to the development of many related schemes (e.g., Qiu et al., 2007; Tian and Feng, 2015; Tian et al., 2008; Tian et al., 2011; Wang et al., 2010; Wang et al., 2008; Wang et al., 2013; Zhang et al., 2022 a; Zhang et al., 2019). These schemes have gradually become mainstream choices in both research and operational applications (Lorenc and Jardak, 2018).

Compared to traditional ensemble-variational hybrid assimilation schemes, the Singular Value Decomposition-three Dimensional Ensemble Variational (SVD-3DEnVar) method (Qiu et al., 2007) is characterized by using singular value decomposition to implicitly represent the covariance relationship between observation increments and background errors, thereby avoiding the computational and storage burdens associated with directly updating the background error covariance matrix. In early implementations of SVD-3DEnVar (e.g., Shao et al., 2009; Zhang et al., 2009), the method directly performed eigenvector decomposition on the entire background field, which caused observations to also adjust distant parts of the background field. For real weather forecasting models with higher dimensions and fewer ensemble samples, such distant correlations are often unreliable and need to be removed using localization techniques. Xu et al. (2011a, b, 2012) localized the SVD-3DEnVar method by employing a local patch assimilation approach and introduced a Gaussian function to further process the observational data. Although this localization strategy enabled successful assimilation of radar data in WRF (Weather Research and Forecasting)-based experiments, applying it to operational systems still faces challenges in computational efficiency.

The main factors limiting the computational efficiency of the SVD-3DEnVar scheme lie in the initial perturbation generation and parallel implementation. Since each assimilation cycle of the SVD-3DEnVar method only assimilates the background

https://doi.org/10.5194/egusphere-2025-4632 Preprint. Discussion started: 14 November 2025

© Author(s) 2025. CC BY 4.0 License.

field once, new ensemble members must be generated by perturbing the background field before each new cycle (refer to Section 3.1). The original ensemble perturbation generation scheme (Evensen, 1994) adopted in SVD-3DEnVar could only support square two-dimensional data, with the constraint that the number of grid points in a single direction must be odd, necessitating additional transformation to overlay with the model background field. This extra processing increased the time consumed in each assimilation cycle. Regarding the assimilation operation, the use of a serial approach to sequentially assimilate each local patch resulted in extremely low efficiency for the SVD-3DEnVar scheme.

To address these issues, this paper aims to optimize the SVD-3DEnVar scheme and improve its efficiency. The performance of the optimized SVD-3DEnVar is then tested by applying to the Tropical Regional Atmosphere Model System (TRAMS), aiming to explore its potential application value in operational typhoon numerical forecasting system. This paper is organized as follows: Section 2 introduces the models and data used in the study; Section 3 presents the SVD-3DEnVar scheme and related technical improvements; Section 4 discusses idealized and real-case experimental results based on the TRAMS model; and Section 5 provides conclusions and discussions.

## 2 Model and data

The TRAMS version 3.0 (Xu et al., 2020) is used for data assimilation and numerical simulation experiments in this study. The model employs a semi-implicit, semi-Lagrangian method for time integration. The model prognostic variables include the three-dimensional wind field (u, v, w), potential temperature (θ), water vapor mixing ratio (q), and dimensionless pressure (π), which are distributed on the horizontal and vertical axes using Arakawa C grids and Charney-Phillips grids, respectively. The physics schemes include the scale-aware New Simplified Arakawa-Schubert (NSAS) cumulus parameterization scheme (Han and Pan, 2011), the WRF single-moment 6-class (WSM6) microphysical schem (Hong et al., 2004), the NCEP (National Centers for Environmental Prediction) Medium-Range Forecast (NMRF) planetary boundary layer scheme (Han and Pan, 2006), the RRTMG long-wave and short-wave radiation scheme (Iacono et al., 2008), and the Slab land-surface model (Dudhia, 1996).

The simulation region is selected based on the area used for operational forecasting (as shown in Figure 1), with a horizontal resolution of 0.09°. The vertical layering consists of 65 layers, using terrain-following coordinates, with the model top at approximately 31 km.

90



Figure 1: Domain of the TRAMS model, with shaded areas representing terrain height (unit: km).

ERA5 reanalysis data (Hersbach et al., 2020) is used to construct the initial and boundary conditions for the TRAMS model. For the real-data assimilation experiment (Section 4.2), sea surface wind field products are derived from multi-source satellite data, including FY3E-WRAD, HY-2B/C/D-SCAT, AMSR2, SMAP, and MWRI. The product has a resolution of 0.25° and a time interval of 6 hours. It can effectively capture winds above 30 m/s, with a root mean square error of approximately 1.5 m/s when compared with in-situ ocean observations. Since August 2024, it has been displayed in real time on the operational website of the Guangdong Meteorological Bureau, providing valuable reference for typhoon monitoring and early warning. The typhoon track and intensity observations used for evaluation come from the best-track dataset provided by the China Meteorological Administration's Tropical Cyclone Data Center (Lu et al., 2021; Ying et al., 2014).

# 3 Method




The operational process of the SVD-3DEnVar assimilation scheme includes three main steps: (1) Initialization step (Figure 3a): The model is initialized to generate the initial field (input) and boundary conditions (bdy). Based on this, the background forecast required for the assimilation process is obtained through direct forecasting. (2) Ensemble perturbation step (Figure 3b): Perturbations are added to the initial field (the perturbed initial field is denoted as input\*) and combined with the boundary conditions (bdy), resulting in a set of ensemble forecast products (ensemble forecast). (3) Assimilation step (Figure 3c): The observational data (observation) undergo a preprocessing procedure, and valid data are selected according to program input requirements after quality control. Using the preprocessed observation data, ensemble forecast results, and background field, the data assimilation calculation is executed to produce a new analysis field (input da). This

analysis field is used as the initial field for the next forecast cycle. Since the model's start time has changed, the boundary data file (bdy up) must be updated synchronously before integration begins.

Figure 2: Flowchart of the SVD-3DEnVar assimilation scheme: (a) Step1: Initialization; (b) Step 2: Ensemble perturbation; (c) Step 3: Assimilation. The pink parallelograms represent the computational steps included in the assimilation scheme, while the yellow squares represent the corresponding data files.

The basic principle of the SVD-3DEnVar scheme is introduced below, followed by an explanation of the optimizations in perturbation construction and parallel operation in this study.

## 3.1 Overview of SVD-3DEnVar


The assimilation process is set at time  $t_0$ , where M initial perturbation fields are added to the background field at  $t_0 - \tau$  to generate M forecast samples, denoted as ensemble  $u_i$  (i = 1, 2, ..., M), where  $\tau$  is the integration time length. These forecast samples are integrated to the  $t_0$  time. At  $t_0$ , a background field without any perturbation, denoted as  $u_b$ , is also provided. The forecast perturbation fields  $\Delta u_i$  (i = 1, 2, ..., M) are obtained by subtracting the background field from the forecast samples:

$$\Delta u_i = u_i - u_b, i = 1, 2, ..., M, \tag{1}$$

Then, the observation perturbation fields  $\Delta d_i$  (i=1,2,...,M) are calculated using the observation operator H. The m-th forecast perturbation field and the observation perturbation field are combined into a column vector:

$$a_m = (\Delta u_m^T, \Delta d_m^T)^T, \tag{2}$$

These column vectors are assembled into a matrix, denoted as A:

$$A = (a_1, a_2, ..., a_M)$$
, (3)

Matrix A is subjected to singular value decomposition:

$$A = B\Lambda V^T \,, \tag{4}$$

where  $\Lambda$  is a diagonal matrix consisting of eigenvalues of A ( $\lambda_1 \ge \lambda_2 \ge \cdots \ge \lambda_M \ge 0$ ) arranged in descending order. According to Equation (2), matrix B of left eigenvalues of A is portioned into two parts, corresponding to the non-zero eigenvalues:

$$b_m = \left(b_m^{uT}, b_m^{dT}\right)^T, \tag{5}$$

where  $b_m^u$  and  $b_m^d$  belong to the model variable space and the observation variable space, respectively.

Let  $x = (\Delta u^T, \Delta d^T)^T$  be the vector to be expanded in terms of the eigenvectors:

$$x = \sum_{r=1}^{K} \alpha_r b_r = b\alpha , \qquad (6)$$

where K is the truncation order. From this, the following relations can be derived:

$$\Delta u = \sum_{r=1}^{K} \alpha_r b_r^u = b^u \alpha \,, \tag{7}$$

and



$$\Delta d = \sum_{r=1}^{K} \alpha_r b_r^d = b^d \alpha \,, \tag{8}$$

The incremental form of the 3DVar objective function is as follows:

$$J(\Delta u) = \Delta u^T P^{-1} \Delta u + (H \Delta u - \Delta y)^T O^{-1} (H \Delta u - \Delta y),$$
(9)

where P is the background error covariance matrix, which, similar to ensemble-based assimilation methods, can be approximated as  $P \approx b^u \Lambda_P^2 (b^u)^T / (M-1)$ ; O is the forecast error covariance, assumed to be a diagonal matrix in this study; y is the observation, with  $\Delta y = y - Hu_b$ . Based on Equations (7) and (8), the objective function (9) can be rewritten as:

$$J(\alpha) = (M - 1)\alpha^{T} \Lambda_{P}^{-2} \alpha + \sum_{r=1}^{K} (\alpha_{r} b_{r}^{d} - \Delta y)^{T} O^{-1} (\alpha_{r} b_{r}^{d} - \Delta y) , \qquad (10)$$

By minimizing the objective function (10), the coefficients  $\alpha$  are obtained, and the required analysis increment  $\Delta u$  is computed using Equation (7).

To prevent any observation point from influencing the global analysis increment, SVD-3DEnVar uses a local patch scheme (Xu et al., 2011a, b, 2012). Specifically, a local block is centered at any model grid point, and a local block with horizontal and vertical radius  $l_h$  and  $l_v$  is selected. SVD-3DEnVar assimilation is then performed within this local block to obtain the analysis field at that grid point. Additionally, a Gaussian weight function is introduced to limit the observational influence within the local block:

$$w(\sigma_h, \sigma_v) = \begin{cases} exp\left(\left(-\frac{r_h^2}{\sigma_h^2}\right) + \left(-\frac{r_v^2}{\sigma_v^2}\right)\right), & (r_h \le l_h \text{ and } r_v \le l_v), \\ 0, & (r_h > l_h \text{ or } r_v \ge l_v) \end{cases}$$

$$(11)$$

https://doi.org/10.5194/egusphere-2025-4632 Preprint. Discussion started: 14 November 2025




where  $r_h$  and  $r_v$  represent the horizontal and vertical distances between the local block center and the observation point, and  $\sigma_h$  and  $\sigma_v$  are the horizontal and vertical localization scale parameters for observations.

## 3.2 Revision of Perturbation Construction Techniques

To address the limitations of the original random perturbation field generation method (such as only supporting square two-dimensional data, requiring odd grid points in a single direction, and the inefficiency of storing 2D data before constructing 3D perturbations), multi-dimensional optimizations have been implemented. These improvements primarily include: breaking the restrictions on grid shape and number to allow direct generation of perturbation fields that match any model grid; optimizing memory usage and efficiency by designing 3D perturbation construction logic that eliminates the need for 2D data read/write steps; and introducing multi-process parallel computing. This section will provide a detailed introduction to these improvements.

The SVD-3DEnVar method uses a Gaussian distribution-based random perturbation field generation approach (Evensen, 1994) when producing ensemble samples. The perturbation field generation process consists of three main steps (Figure 3a): first, a data pool satisfying the specified Gaussian distribution is generated and divided into intervals; second, sampling is conducted from the data pool intervals using spiral enumeration and neighborhood statistical relationships; third, to ensure the perturbation field has spatial scale characteristics, a five-point smoothing is applied to the generated 2D perturbation field, and a vertical weighting combination is used to construct the 3D perturbation field.

In the first step, the original scheme pre-generates a sample data pool by specifying the number of sample intervals and the total number of samples (usually a very large quantity). This approach not only occupies an enormous amount of memory but also results in relatively low efficiency in subsequent sampling. In the second step, the original scheme is limited to square grids and odd grid points, which often do not match the grids used in the model. Additionally, the original method first generates a sufficient number of 2D perturbation fields and stores them locally before constructing the 3D perturbation field. Prior to the 3D perturbation field construction, the 2D perturbation fields are read sequentially from local storage. Since the grids typically do not match, the perturbation fields require 2D interpolation to match the model grid. This approach has several drawbacks: on the one hand, the generation of 2D perturbation fields is indirect, and the interpolation may distort the original statistical characteristics of the perturbations; on the other hand, the additional read/write steps not only impact efficiency but also occupy local storage space.



Figure 3: (a) Schematic of the optimization process for random perturbation generation in the SVD-3DEnVar scheme; (b) shows the horizontal distribution of the generated perturbation field; (c) shows the vertical cross-section taken along the black dashed line in (b).

In light of this, optimizations are made as following: First, the grid limitation issue is addressed. As shown in the blue box of Figure 3a, given any grid, the relative center point is selected, and spiral sampling begins. If the current point is invalid, it is skipped, and sampling continues until all points are sampled. Meanwhile, it is found that the practice of pre-allocating a sample pool in the first step leads to low efficiency in disturbance field generation. To address this issue, we consider eliminating this step; instead, in the alternative scheme, during sampling, the truncated normal distribution is calculated directly based on neighborhood statistical information, thereby quickly obtaining values that meet the requirements at a relatively low cost. Furthermore, to overcome the inefficiency of the original program—where 2D perturbation fields are first generated, stored locally, and later read and interpolated to form 3D fields—the process is streamlined. Now, the required number of 3D perturbation fields, along with the number of grid points in the horizontal, and vertical directions, are directly specified. The 2D data is immediately used in 3D construction, and once completed, the 3D perturbation file is output in one step. This optimization significantly improves efficiency by eliminating redundant read/write operations, storage and sampling of large sample arrays.

Furthermore, to further enhance computational efficiency, the sample generation process supports multi-process parallelization. The specific strategy is as follows: for each 3D field, the corresponding number of 2D perturbation fields is





generated in parallel according to the number of layers. These are then merged and output as the 3D field file, and the same process is repeated for the next 3D field. This approach limits memory usage by controlling the parallelization scope to avoid overflow, and it allows flexible adjustment of the parallelization scale for the 2D data within each 3D field to accommodate different computing resources. Ultimately, this improves performance while maintaining memory stability and resource utilization efficiency.

According to the original scheme, generating a 3D perturbation field with a grid dimension of  $101\times101\times67$  took approximately 1340 seconds (over 22 minutes). Note that this time is already close to the duration required for running the SVD-3DEnVar assimilation step (refer to Figure 5a), making it clearly unacceptable in operational applications. Without enabling 2D field parallel computing, the optimized scheme completes the same task in just 2.2 seconds, achieving a computational efficiency improvement of approximately 600 times. For the TRAMS model with a grid dimension of  $883\times553\times67$ , further efficiency gains can be achieved by activating parallel computing for the 2D fields. Figures 3b and 3c show the horizontal and vertical distributions of a randomly selected 3D perturbation field, with characteristics that are consistent with the results of Xu et al. (2011b).

## 3.3 Parallel implementation

The original SVD-3DEnVar data assimilation program performs calculations serially for each local patch corresponding to each grid, and the computation time is insufficient to meet the demands of large-domain model assimilation operations in practical applications. Therefore, a parallelization scheme is required. This study draws inspiration from the LETKF scheme (Miyoshi and Yamane, 2007) and combines patch parallel computing with load balancing strategies. A "grid partitioning global marking - dimensionality reduction assignment" strategy is adopted to achieve high operational efficiency. At the same time, "node root process pointer sharing" and "local data processing" techniques are employed to optimize data I/O speed and memory management, reducing redundant computations and storage usage.

The specific process is shown in Figure 4. First, the model grid is partitioned into multiple parallel regions according to the number of processes (marked as red boxes). Each process checks each grid point within its assigned region to determine whether it satisfies the assimilation trigger conditions (i.e., whether the number of observations reaches the preset threshold).

The grid points that meet the conditions are marked, and after all grid points are checked, global marking information is collected. To reduce unnecessary computational overhead, only the location information of the observation data is used for preliminary judgment and filtering in this stage, avoiding redundant calculations. The filtered model grid points for assimilation are then reduced in dimension to generate corresponding coordinate sequences  $(x_1, y_1), (x_2, y_2), \dots, (x_D, y_D)$ , which are then rebalanced and redistributed according to the number of processes. For example, in Figure 4, the coordinate sequence  $(x_1, y_1), (x_2, y_2), \dots, (x_d, y_d)$  is assigned to node  $N_1$ .

Additionally, to avoid memory overflow on a single node due to excessive process numbers, this study synchronously optimizes the program's memory management mechanism. First, the data required for assimilation (including observations, ensemble samples, and background fields) are stored in the root process of each node (for example, the root process





corresponding to node  $N_1$  is labeled  $P_1$  in Figure 4). Other processes within the node (such as  $P_2$ ) then retrieve data from the root process through pointer sharing. The "root process pointer sharing" strategy ensures that large arrays are not redundantly copied, preventing storage space redundancy. At the same time, in local assimilation calculations, only locally scoped data are used for computations, avoiding the processing of global data. Through the dual approach of "spatial restriction + pointer sharing", both memory usage and computational efficiency are optimized in parallel.

Figure 4: Schematic of the parallel mechanism in the SVD-3DEnVar assimilation scheme. The black grid represents the model's horizontal grid, the orange grid divisions represent individual parallel domains, and the solid circle scatter points represent the observation data locations.  $(x_i, y_i)$  denotes the i-th (i=1,2,...,m) horizontal grid point to be assimilated,  $N_j$  represents the j-th (j=1,2,...,J) node, and  $P_1$ ,  $P_i$ , and  $P_j$  represent the root processes corresponding to nodes  $N_1$ ,  $N_j$ , and  $N_j$ , respectively. "Data" refers to the data required for assimilation (including observations, ensemble samples, and background fields).

It should be noted that if the model region is directly partitioned into grids, local areas may have insufficient observation counts, causing early termination of assimilation in those regions while other processes continue running. This leads to idle processes and underutilized computational resources, which becomes a significant bottleneck in efficiency improvement. The approach adopted in this study allows for synchronous invocation of all processes, and each process completes the assimilation task around the same time, effectively avoiding the problem of idle waiting in a single process, thus significantly improving resource utilization.

To evaluate the impact of the parallel strategy on the operational efficiency of the SVD-3DEnVar assimilation scheme, targeted test experiments were designed in this study. First, the local block radius was fixed at 10 times the grid size, and only the number of nodes was varied, with all other experimental conditions remaining identical to those in the ideal experiment described in Section 4.1. Figure 5a showed the sensitivity of the assimilation computation time to the number of nodes. In the experiment, the number of nodes increased from 1 to 300, with each node equipped with 64 CPUs. The time spent on each key component of the assimilation scheme was then recorded. As the number of computing nodes increased from 1 to 150, the computation time for the assimilation program decreased linearly. When the number of nodes exceeded





265 150, the parallel efficiency of the assimilation program reached saturation, and the required time stabilized around 15 minutes.

Specifically, the computation time for the assimilation program was primarily distributed across three core steps: data I/O, resource allocation, and assimilation computation. In terms of data writing, the time spent remained stable with no significant fluctuation. The reason for this is that data writing is only performed by the root process and is not affected by the increase in the number of nodes. In the resource allocation step, due to the secondary allocation strategy, the time spent during the first allocation phase showed clear dependence on the number of nodes. When the number of nodes was small, the time spent on horizontal statistical processing of the assimilation grid points was relatively high. However, once the number of nodes reached a certain threshold, this time was significantly reduced and eventually approached zero. The data reading step showed a slight increase in time as the number of nodes increased. This phenomenon is likely related to data contention: multiple nodes simultaneously make read requests for the same data, causing data access conflicts and leading to a slight increase in time. This is a common phenomenon in parallel computing and is within a reasonable range. In contrast, the time spent on the assimilation computation step generally showed a downward trend. Notably, between 170 and 210 nodes, the time spent on the assimilation computation increased abnormally. The possible reason for this is that the actual time spent on assimilation computation can vary depending on the location of observations and the characteristics of the data (e.g., due to differences in the number of iterations). When the number of nodes was in this range, the task assignment for the experimental dataset did not achieve "efficiency uniformity," and some processes were assigned tasks with relatively high computational loads, which increased the overall time. However, since the current task assignment strategy only ensures an even distribution of the number of tasks, without considering differences in the computational time of individual tasks, this anomaly has a limited impact on the overall efficiency in practical applications and can be disregarded.

Since SVD-3DEnVar employs a local patch parallel strategy, the size of each local patch is a key factor in determining the assimilation computation efficiency. Figure 5b tests the assimilation time for different horizontal localization radii with 150 nodes. The results shows that as the localization radius increases, the computation time for SVD-3DEnVar also increases linearly. This increase primarily comes from the assimilation computation step.

In summary, the parallel assimilation strategy designed in this study demonstrates good load balancing performance, which makes the SVD-3DEnVar scheme suitable for practical operational applications.

Figure 5: Computation time of the SVD-3DEnVar scheme under different numbers of nodes (a) and local patch radii (b). The specific design of the test experiment can be found in section 4.1 of the OSSE simulation experiment. The gray dashed line denotes the computational efficiency when using 150 nodes and a localization radius equal to 10 times the grid spacing, which represents a parallel parameter configuration well-suited for operational applications.

## 4 Experimental Design and Results Analysis

## 4.1 Experimental Design

295

305

300 To test the performance of the revised SVD-3DEnVar scheme, two sets of assimilation experiments were conducted based on the TRAMS model in this section.

The first group of experiments was the Observing System Simulation Experiment (OSSE) for Typhoon Khanun (2023, Typhoon No. 6) under ideal conditions, as shown in Figure 6a. The control experiment (CTL) was directly integrated forward from 12:00 UTC on July 28, while the assimilation (DA) experiment had the analysis time at 00:00 UTC on July 29. The background field for the DA experiment was derived from a 12-hour forward integration starting at 12:00 UTC on July 28, and the true field was obtained from a 24-hour forward integration starting at 00:00 UTC on July 28. The horizontal uwind component in the vertical layers 8 to 32 from the true field was used as the observation. Considering model resolution, observation data, and parallel efficiency, the radius of the local block was set to 10 times the grid size, and the observation localization scale parameters were set to 3 times the grid size (i.e.,  $l_h = l_v = 10$ ,  $\sigma_h = \sigma_v = 3$ ). The model variables updated



in the assimilation process include the v-wind field (v), potential temperature ( $\theta$ ), dimensionless pressure ( $\pi$ ), and water vapor mixing ratio (q), making a total of 5 variables. The experiment used an ensemble of 30 members, with a truncation order of K = 27.

Figure 6: Assimilation experiment flow design: (a) OSSE experiment; (b) Sea surface wind assimilation experiment. The horizontal axis represents the corresponding time. CTL refers to the control experiment without any assimilation, and the specific assimilation designs for DA, DA1, DA2, and DA3 are described in the relevant text.

Following the validation of the SVD-3DEnVar assimilation scheme's effectiveness in the OSSE experiment, further assimilation tests were conducted using actual observational data. The testing period was chosen to coincide with Typhoon "Mocha" (2024, Typhoon No. 11), and the assimilated observational data were the satellite-fusion sea surface wind field inversion products described in Section 2 (with an observation height of 10m). The goal was to preliminarily assess the performance of the scheme in a real-world weather process. It should be noted that the reason why the same typhoon case was not selected for both the OSSE and the actual assimilation experiment is that the sea surface wind product has only been made publicly available since September 2024. The experimental design is shown in Figure 6b.

The control experiment (CTL) was obtained by directly integrating the global analysis field from 12:00 UTC on September 5, 2024. The first assimilation was initiated at 00:00 UTC on September 6, 2024, with the background field being derived from a 12-hour forecast starting at 12:00 UTC on September 5, 2024. To further test the impact of multiple-cycle assimilation on model forecasting in an operational environment, after the first assimilation (denoted as DA1), two consecutive assimilation experiments were designed (denoted as DA2 and DA3), with a 6-hour interval between each assimilation cycle. The analysis variables, localization parameters, number of ensemble members, and truncation order for this experiment were identical to those used in the previous OSSE experiment.



## 4.2 Results of OSSE Experiment

This section primarily verifies the rationality of the SVD-3DEnVar assimilation scheme in an idealized typhoon forecasting scenario. The "true error" used in the experiment is obtained by subtracting the pre-defined "true field" from the forecast field (refer to Figure 6a).

The analysis increments and true errors of the u-wind field at the 10th model level (approximately at 1.5 km height) are first selected to compare the consistency of the horizontal distribution. The analysis increment of the u-wind field and the true error show very similar spatial distribution and magnitude (Figures 7a-b). This is because in the OSSE experiment, observational data for the u-wind field on this model level were directly provided. In contrast, for the other four variables (v,  $\theta$ ,  $\pi$ , q) where no direct observations were provided, the analysis increments do not match the true error as closely as the u-wind field (Figures 7c-j). However, the analysis increments still reasonably capture the spatial distribution of the errors, particularly in the typhoon region.

This indicates that the SVD-3DEnVar assimilation scheme can, even when assimilating only a single variable, adjust the other variables indirectly through the flow-dependent background error covariance relationships contained in the ensemble samples, thus producing an initial analysis field that is physically coherent.

Figure 7: The true errors (first column) and analysis increments (second column) at the model's 10th layer (approximately 1.5 km altitude). From top to bottom, the variables correspond to u (a, b), v (c, d),  $\theta$  (e, f),  $\pi$  (g, h), and q (i, j).

Further comparisons are made between the typhoon track and intensity before and after assimilation, using observations from the true field. The CTL experiment forecast shows a systematic northward bias in the typhoon track (Figure 8a). For typhoon intensity, the CTL experiment exhibits an overestimation of intensity between 24-72 hours and an underestimation

between 114-144 hours (Figure 8b). The DA experiment effectively reduces the forecast bias in both the typhoon track and intensity. These results indicate that after assimilating the wind field, the SVD-3DEnVar scheme can significantly improve typhoon track and intensity forecasts. This improvement is, on one hand, due to a more accurate description of the dynamical and thermo-dynamical structure of the initial typhoon core, and on the other hand, is also related to the reduction of initial errors in the large-scale steering flow through data assimilation.


Figure 8: OSSE simulation results for Typhoon Khanun under ideal conditions: (a) typhoon track and (b) minimal sea-level pressure (unit: hPa).

## 365 4.3 Results of OSSE Experiment

This section further tests the impact of sea surface wind field data assimilation based on SVD-3DEnVar on typhoon forecasting under real-world conditions. First, the deviation characteristics between the available observational data and model forecast results are compared. It is found that at the analysis time (00:00 UTC on September 6, 2024), in the typhoon-affected area, the model forecasted wind speeds are generally lower than the observed wind speeds (as shown in Figure 9). This is likely due to the weaker intensity representation of super typhoons in the ERA5 reanalysis data, which serves as the driving field for the model (Li et al., 2024).



Figure 9: Sea surface wind background field (unit: m/s) at 00:00 UTC on September 6, 2024 (analysis time) (a), observed field (b), and their differences (c).

After assimilating the sea surface wind field observational data using SVD-3DEnVar, the model effectively adjusts the previously weak typhoon circulation (Figure 10a). The central pressure of the typhoon decreases as the circulation strengthens (Figure 10c), which is consistent with the gradient wind balance relationship. In terms of the thermodynamic structure, the temperature in the typhoon core increases after adjustment, indicating a more pronounced warm-core structure (Figure 10e), while the water vapor increases (Figure 10g). This is due to the enhanced suction effect as wind speed increases, which leads to more water vapor from the sea surface being pumped into the typhoon center, causing an increase in specific humidity in the surrounding region. This also contributes to the subsequent intensification of the typhoon forecast. From the vertical profile, the impact of assimilating the sea surface wind field is mainly confined to the boundary layer below 1 km (Figures 10b, d, f, h). Overall, the incremental distribution of the four key meteorological elements—wind speed, pressure, potential temperature, and specific humidity—after assimilation is consistent with the physical laws and expected adjustments during the typhoon intensification process. This confirms the rationality of assimilating observational data from the perspective of the synergistic evolution of these variables and further validates the reliability of the SVD-3DEnVar scheme in the assimilation application for actual typhoon events.

Figure 10: Assimilation analysis increments at the model's second layer (the first column) and the vertical cross-section along the black line in the first column (the second column). From top to bottom, the corresponding variables are u (a, b), v (c, d),  $\theta$  (e, f),  $\pi$  (g, h), and q (i, j).


Building on the validation of the analysis increment's rationality, further tests were conducted to assess the impact of sea surface wind field data assimilation on typhoon track and intensity forecasts. Three assimilation experiments—DA1, DA2, and DA3—were set up, with the specific details of these experiments provided in Section 4.1.



As shown in Figure 11a, the assimilation experiments have little impact on the typhoon's track. This is primarily because the typhoon's movement is mainly controlled by the large-scale steering flow at the edge of the subtropical high, while the increments from data assimilation (refer to Figure 10) are mostly concentrated in the typhoon's vortex region, with vertical influence confined to the boundary layer. Assimilating the sea surface wind field effectively corrects the underestimation of the initial typhoon intensity in the CTL experiment (Figure 11b). However, this correction cannot be maintained, with the forecast returning to the pre-assimilation levels after approximately 6 hours forecast. This suggests that adjusting only the low-level typhoon structure is insufficient to substantially improve typhoon intensity forecast biases. Further improvements require combining typhoon initialization (e.g., Zhang et al., 2026) and the assimilation of additional observational data (e.g., cloud steering winds) (e.g., Li et al., 2015) to adjust the middle and upper-level typhoon structures. As the assimilation cycles are conducted at 6-hour intervals, the results of DA2 and DA3 show little difference from DA1.

Figure 11: Typhoon Yagi path (a) and 10m maximum wind speed (unit: m/s) (b) before and after assimilation.






## 5 Conclusion and Discussion

To further promote the SVD-3DEnVar assimilation scheme toward practical operational applications, this study proposes an efficient 3D perturbation generation scheme and a local patch parallel assimilation strategy. The OSSE test results based on the TRAMS model show that after computational efficiency optimization, the time required to generate a 3D perturbation field has been reduced from 22 minutes to 2.2 seconds, while the runtime of the assimilation process for the SVD-3DEnVar scheme has decreased from 1,700 minutes under serial execution to less than 15 minutes (using parallel processing across 150 nodes). Additionally, the reasonableness of optimized SVD-3DEnVar assimilation scheme was preliminarily verified through idealized experiments (Typhoon Khanun, 2023) and real-world experiments (Typhoon Yagi, 2024).

The experimental results in Figure 11 indicate that assimilating only sea surface wind data is not sufficient to effectively improve the model's typhoon forecasts. To obtain a more accurate initial field, it is necessary to assimilate as much observational data as possible from the middle and upper atmosphere and other forecast variables. Therefore, future work should focus on introducing additional types of observational operators and corresponding quality control systems into the SVD-3DEnVar scheme to meet the multi-source observational data assimilation needs in operational forecasts.

It is noted that the analysis increments from SVD-3DEnVar often contain small-scale noise, especially for variables that lack direct observations (see Figures 7 and 10). This noise can generate spurious gravity waves in the model, which may affect the subsequent forecast quality, particularly in the case of cycling assimilation. This could be due to the use of a single localization scale in the SVD-3DEnVar scheme, as the background error covariance relationships between different analysis variables are embedded in the singular vectors of the local ensemble samples, with the scale characteristics primarily dependent on the localization radius and observation localization scale parameters (as discussed in Section 3.1).

To address these issues, we plan to focus on two aspects. First, we will adopt the Incremental Analysis Update (IAU) technique (Bloom et al., 1996) to alleviate the impact of initial field adjustments on the model. The IAU technique decomposes the analysis increments into smaller components and gradually incorporates them into the model forecast, effectively filtering out short-wavelength noise from the increments, thereby improving the consistency between the model and initial conditions. We have already achieved good practical results using the IAU method in radar data assimilation (Lin et al., 2025). On the other hand, multi-scale assimilation techniques are also an effective way to improve the coordination of increments across different scales. For example, Zhang and Tian (2018) implemented a multi-scale assimilation scheme based on a multi-grid approach (Xie et al., 2011) in their NLS-4DVar method. Given the similarities between this method and SVD-3DEnVar (such as both representing analysis increments via a linear combination of ensemble perturbations), their research provides valuable insights for the future development of multi-scale assimilation in the SVD-3DEnVar framework. With the rapid development of artificial intelligence (AI) and machine learning (ML) technologies, their application in meteorological data assimilation shows promising prospects. For instance, the machine learning-based data assimilation method DiffDA utilizes a pre-trained GraphCast weather forecasting model as a denoising diffusion model and combines

predicted states with sparse observational data to assimilate atmospheric variable (Huang et al., 2024). FuXiDA, based on a

https://doi.org/10.5194/egusphere-2025-4632 Preprint. Discussion started: 14 November 2025

© Author(s) 2025. CC BY 4.0 License.






unified fusion neural network, effectively adjusts the weights between observations and background without the need to estimate error covariance matrices. This deep learning assimilation framework, when applied to satellite observations, not only reduces analysis errors but also significantly improves forecast performance (Xu et al., 2025b). ADAF also approximates the product of the Kalman gain matrix and the innovation vector using neural networks, replacing traditional assimilation frameworks, and is able to rapidly process large amounts of observations to generate high-quality analysis fields at low computational cost (Xiang et al., 2025). Keller and Potthast (2024) proposed an AI-based variational assimilation method (AI-Var), where a trained neural network minimizes the cost function in the variational process to obtain an accurate initial field needed for numerical forecasts. The further integration of machine learning with the SVD-3DEnVar scheme to address challenges such as complex nonlinear observation operator design and multi-scale assimilation is a promising direction for future research.

*Data availability*. The simulated and assimilated results, observation data, and plotting scripts can be obtained at https://doi.org/10.57760/sciencedb.28464 (Liu and Xu, 2025b).

460 Code availability. The source code for the TRAMS 3.0 model and the SVD-3DEnVar data assimilation framework can be obtained at https://doi.org/10.57760/sciencedb.30837 (Liu and Xu, 2025a).

Author contribution. KL designed and implemented the data assimilation system, ran all tests and simulations, created visualizations and validation analyses, carried out the formal analysis, and produced the original draft. DX shaped the conceptualization and methodology, shared reference source code from an earlier version, led the investigation efforts, refined the original draft, and obtained funding with MZ. FZ and JH provided computational resources; separately, JH offered guidance on code-related questions, while FZ provided partial theoretical guidance. JCL and MZ critiqued and polished the manuscript. QH and YZ supplied multi-source satellite data for real data assimilation experiments. DZ, YL, and CL oversaw the project. DX and BZ managed the project administration.

Competing interests. The authors declare that they have no conflict of interest.

Acknowledgments. We thank for the technical support of the National large Scientific and Technological Infrastructure "Earth System Numerical Simulation Facility" (https://cstr.cn/31134.02.EL).

*Financial support*. This work was supported by the National Natural Science Foundation of China (Grant Nos. U2142213, 42405162).

Soc., 146, 1999–2049, https://doi.org/10.1002/qj.3803, 2020.

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
