# Peer review of "Improvement of the Computational Efficiency in SVD-3DEnVar Data Assimilation Scheme and Its Preliminary Application to the TRAMS 3.0 Model"

_EGUsphere, 2025_

## Author Comment (AC1)

**Response to Reviewer 1**

We sincerely thank the reviewer for the insightful comments and constructive suggestions, which have significantly helped us improve the quality of our manuscript. We have carefully considered all points and have revised the manuscript accordingly. Below, we provide a point-by-point response to the comments.

**Comment 1:**

**Regarding the choice of methodology, it is recommended that you explain why SVD-4DVar was not adopted in favor of SVD-3DVar, while briefly analyzing the core challenges of the latter.**

Response:

We appreciate the reviewer's suggestion. The decision to adopt SVD-3DEnVar instead of SVD-4DEnVar was primarily motivated by two factors:

Computational Efficiency and Operational Feasibility: The main objective of this study is to improve computational efficiency for operational typhoon forecasting, so the relatively simple 3DVar scheme is chosen for experimentation...

Observation Frequency: The sea surface wind observations used in our real-data experiments are available only every 6 hours. This low temporal resolution does not fully leverage the temporal continuity advantages of 4DEnVar.

We acknowledge that SVD-3DEnVar has its own challenges, which we now discuss in the revised manuscript (Section 5). The primary limitations include: The use of a single localization scale, which may not optimally handle multi-scale observations (e.g., surface, satellite, and radar); The limited ability of linear combinations of singular vectors to represent strongly nonlinear relationships, especially for unobserved variables. These challenges will guide our future work on multi-scale assimilation and machine-learning-enhanced methods.

**Comment 2:**

**It should be noted that the singular value decomposition (SVD) of matrix A is extremely challenging in practice due to its large dimensions ($N_x + N_y$), posing significant difficulties in terms of both storage and computation. A discussion on this aspect is recommended.**

Response:

We fully agree with the reviewer. To address the computational and storage challenges of

performing SVD on the large matrix A, we implemented a local patch assimilation strategy. This approach significantly reduces the effective dimensions of A by horizontal and vertical localization. Only observations within a specified horizontal ($l_h$) and vertical ($l_v$) radius from the central grid point are included. As a result, both $N_x$ (model variables in the local patch) and $N_y$ (observations within the local patch) are drastically reduced. This makes the SVD computationally feasible without sacrificing the flow-dependent covariance information. We have added a clarification in Section 3.1 to explain this strategy.

**Comment 3:**

**While equation (11) provides the Gaussian weight function, the localization scheme used in SVD-3DVar should be presented in more detail to enhance the completeness of the paper.**

Response:

Thank you for this suggestion. We have expanded the description of the localization scheme in Section 3.1. The revised text now reads:

"The Gaussian weight function defined in Equation (11) is applied to each observation within the local patch. The horizontal and vertical localization scales ($\sigma_h$ and $\sigma_v$) control the rate at which observation influence decays with distance. This localization ensures that only observations within a specified radius significantly impact the analysis increment at the center of the local patch, thereby mitigating spurious long-range correlations and improving the stability and accuracy of the assimilation."

**Comment 4:**

**In recent years, 4DEnVar methods have advanced rapidly. To reflect an up-to-date understanding of the field, it is advisable to include references to relevant studies published between 2022 and 2025.**

Response:

We thank the reviewer for this suggestion. We have now incorporated several recent references on 4DEnVar advancements in the Introduction and Conclusion sections, including:

Inverarity et al. (2023) on hybrid En-4DEnVar in the Met Office system;

Berre and Arbogast (2024) on hybrid covariances at Météo-France;

Lu and Wang (2024) on scale-dependent localization in hurricane forecasting;

Thiruvengadam and Wang (2025) on convective-scale 4DEnVar;

Wang et al. (2025) on CubeSat radiance assimilation.

These additions help contextualize our work within the evolving landscape of ensemble-variational methods.

**Comment 5:**

**Regarding the generation of initial samples, several classical works (e.g. those by Evensen) have achieved high memory efficiency. It would be beneficial to reference these works and discuss their relevance to the present method. From a practical perspective, the main computational burden in parallelization typically lies in the ensemble forecast component, which should also be addressed.**

Response:

We agree with the reviewer. The initial perturbation generation in SVD-3DEnVar is based on the Gaussian random field method introduced by Evensen (1994), which is known for its memory efficiency and statistical robustness. However, the original implementation was limited to 2D square grids with odd numbers of points, which motivated our multi-dimensional and parallel optimizations. We have mentioned Evensen's foundational work and clarify how our optimizations build upon it.

Regarding the computational burden of ensemble forecasts: yes, the ensemble integration is indeed the most computationally intensive part in parallel implementations. However, since ensemble members are independent, they can be run concurrently if sufficient computational resources are available. While this study focuses on optimizing the perturbation generation and assimilation steps, we acknowledge the resource demands of ensemble forecasting and will address this in future work.

**Comment 6:**

**As this is an ensemble-based method, it is recommended that the ensemble sample update strategy in SVD-3DVar is explained briefly to improve the completeness of the methodological description.**

Response:

We have added the following explanation to Section 1:

"Unlike traditional ensemble assimilation schemes (such as EnKF), which directly update each ensemble member during cyclic assimilation, SVD-3DEnVar only assimilates and updates the control forecast. Therefore, after each assimilation cycle, the updated analysis field must be perturbed again to generate the initial conditions for the next cycle's ensemble forecast. This approach has the advantage of avoiding filter divergence and, in the presence of model errors, performs better than EnKF (Qiu et al., 2007)."

---

## Author Comment (AC3)

**Response to Reviewer 3**

We sincerely thank the reviewer for the thorough evaluation and insightful comments, which have helped us improve the manuscript. Below, we address each point raised.

**Comment 1:Clarification on Parallelization and Load Balancing (Line 255)**

**The authors state: "The approach adopted in this study allows for synchronous invocation of all processes... effectively avoiding the problem of idle waiting... thus significantly improving resource utilization."**

**This explanation is scientifically insufficient. "Synchronous invocation" implies starting tasks simultaneously, but it does not inherently solve the computational load balancing problem. In a local patch-based domain decomposition, observations are rarely uniformly distributed. Consequently, using a static domain decomposition inevitably results in some processes handling significantly more observations than others. How does "synchronous invocation" prevent processes with fewer observations from finishing early and idling?**

**The authors seem to imply that a task redistribution mechanism is in place, but it is not clearly described. A detailed explanation of how the workload is balanced across processes is required (e.g., are grid points dynamically redistributed based on observation density or computational cost?).**

Response:

Thank you for highlighting this crucial point. Our parallel strategy involves a two-step process to achieve load balancing.

**Initial Parallel Screening**: The model grid is statically partitioned. Within each partition, all processes work in parallel to screen each grid point. The criterion for marking a point as "to be assimilated" is whether the number of valid observations within its local patch meets or exceeds the preset truncation order (K).

**Dynamic Task Redistribution**: After this global parallel screening, all marked grid points are collected. These points (now a scattered set, not a regular grid) are then evenly redistributed among all available processes. This ensures that each process receives a nearly equal number of assimilation tasks, effectively balancing the computational load.

Through the above two steps, the assimilation computational tasks of SVD-3DEnVar can be essentially evenly distributed across various nodes.

We have added this detailed explanation to Section 3.3 of the manuscript.

**Comment 2: Theoretical Validity of OSSE Results (Figure 7)**

**In Figure 7, the analysis increment for the u-wind component (7b) appears almost identical to the "true error" (7a) in both spatial pattern and magnitude. According to data assimilation theory, the analysis increment is the product of the Kalman gain and the innovation (observation minus background). Unless the observation error (R) was set to zero or the background error covariance (B) was inflated to an unrealistic magnitude, the analysis increment should not match the true error so perfectly. This result raises serious questions about the experimental setup or the plotting (e.g., potential confusion between innovation and increment). The authors must verify this result and provide a physical or theoretical justification.**

Response:

We appreciate the reviewer's careful scrutiny. The close match is expected and validates the experimental design and algorithm behavior under the specific OSSE conditions:

**Dense, Perfect Observations**: In the OSSE, direct observations of the u-wind component are provided at every model grid point within layers 8-32. This creates an exceptionally dense and spatially complete observing network for this variable.

**Methodology**: For computational convenience, in the implementation described in this paper, we did not directly minimize the objective function (10) through iterative calculations. Instead, following the approach of Qiu and Chou (2005), we employed the method of least squares to solve for $\alpha$:

$$\Delta y = \sum_{r=1}^{K} \alpha_r b_r^d \qquad\qquad (11)$$

By solving the algebra equations (11), the coefficients $\alpha$ are obtained, and the required analysis increment $\Delta u$ is computed using Equation (7). In this SVD-3DEnVar implementation, the coefficients $\alpha$ are determined solely from observational information. When sufficient observations are available (i.e., the number of observations exceeds or equals the truncation order K), the system is well-posed, a stable minimum-norm solution can be obtained because the solution is constrained to the low-dimensional subspace spanned by the singular vectors. This eliminates the dependence on background error statistics required in traditional methods and simplifies the assimilation process.

On the other hand, since the SVD method only retains the first K eigenvectors to fit the observation increments, it effectively filters out small-scale observational increment information (corresponding to eigenvectors with relatively small eigenvalues). Therefore, even when random noise is added to the observations in the OSSE experiment, it does not significantly impact the assimilation results. This is one of the differences between the SVD method and traditional assimilation methods, namely its lower sensitivity to random observational errors.

Reference: Qiu, C. and Chou, J.: Four-dimensional data assimilation method based on SVD: Theoretical aspect, Theor. Appl. Climatol., 83, 51–57, https://doi.org/10.1007/s00704-005-0162-z, 2005.

**Comment 3: Ambiguity in Observation Processing (Line 108)**

**The manuscript mentions: "valid data are selected according to program input requirements after quality control."**

**This description is too vague for a research article. How exactly are "valid data" selected? Does this process involve spatial thinning, super-obbing, or specific domain checks? A concrete description of the selection criteria is necessary to ensure reproducibility.**

Response:

We agree that the description was too brief. For the specific experiments presented in this paper:

**OSSE Experiment:** Simulated u-wind observations were used directly at their native model grid points without additional thinning or super-obbing.

**Real-Data Experiment (Sea Surface Winds)**: The multi-source satellite wind product (described in Section 2) underwent its quality control and bias correction during the retrieval process. For ingestion into our SVD-3DEnVar system, we applied a simple background check: observations were rejected if the absolute difference between the observation and the background (mapped to observation space) exceeded a threshold (set to 20 m/s for wind speed). No additional spatial thinning or supero-bbing was applied for these preliminary tests.

The primary focus of this study was computational optimization. Future work incorporating higher-resolution data (e.g., radar) will require and implement more sophisticated preprocessing (e.g., thinning, super-obbing). We will clarify this in Section 4.1.

**Comment 4: Mathematical Rigor and Definitions (Section 3.1)**
**The mathematical description of the SVD-3DEnVar scheme lacks precision:**

**- Line 135: The matrix AA is rectangular; therefore, it has singular values**, not "eigenvalues."

**- Undefined Symbols: In Eq. (4) (A=B$\Lambda$V^T), the term V^T is not defined. Similarly, in Eq. (6) (x=b$\alpha$), the variable $\alpha\alpha$ appears without a proper definition. The authors should explicitly define these variables to aid reader understanding.**

Response:

Thank you for catching these issues. We have corrected them in the manuscript.

"Eigenvalues" has been replaced with "singular values" throughout.

In the text following Equation (4), we have added: "...where $\Lambda$ is a diagonal matrix of singular values..., and B and V are orthogonal matrices containing the left and right singular vectors of A, respectively."

Following Equation (6), we have added: "...where α = (α₁, α₂, ..., α_K)^T is the vector of coefficients to be determined by minimizing the cost function."

**Comment 5: Overgeneralization of Results (Figure 8)**

**Based on a single OSSE case shown in Figure 8, the authors claim that "the SVD-3DEnVar scheme can significantly improve typhoon track and intensity forecasts." This statement is too strong for a single idealized experiment. It would be more appropriate to state that the scheme demonstrates potential for improvement in this specific case.**

Response:

We agree with the reviewer. The statement has been toned down as suggested. The relevant sentence in Section 4.2 now reads:

"The DA experiment effectively reduces the forecast bias in both the typhoon track and intensity for this specific case. These results indicate that after assimilating the wind field, the SVD-3DEnVar scheme demonstrates the potential to improve typhoon track and intensity forecasts in this idealized framework."

**Comment 6: Interpretation of "Operational Feasibility"**

**The manuscript frequently emphasizes the operational applicability of the proposed scheme based on reduced wall-clock time. However, operational feasibility depends not only on speed but also on resource availability and system stability. Given that some results rely on a large number of computing nodes (up to 300 nodes), it would be beneficial for the authors to discuss whether similar performance gains can be achieved under more constrained computational resources, which is a common reality for many operational centers.**

Response:

This is a valuable point. Our scaling test up to 300 nodes was to explore the saturation point of parallelism. As seen in Fig. 5a, the performance gain (reduction in wall-clock time) slows significantly beyond ~150 nodes for our test configuration. More importantly, substantial gains are achieved with far fewer resources.

Using approximately 30 nodes (with 64 cores each), the assimilation time is reduced to about 1 hour, which is already a dramatic improvement from the original serial execution.

The current experiments assimilate very dense data (effectively all grid points in the OSSE). In an operational setting, realistic observations would be much sparser after standard quality control and thinning procedures. This would further reduce the computational cost, potentially allowing runtime targets (e.g., under 30 minutes) to be met with fewer than 60 nodes.

Since operational ensemble prediction systems already require tens to hundreds of nodes to run the ensemble forecasts, dedicating a comparable subset for an efficient assimilation step is feasible and represents a significant step toward operational readiness. We will add a brief discussion on this in Section 5 (Conclusion).

**Comment 7: Technical Corrections and Visualization Issues**

**1. Figure 4: The red boxes representing parallel domains are difficult to distinguish because the model grid points are also represented by lines. I suggest representing the** model grid points as dots **and using** lines only for the red boxes **to improve visual clarity.**

**Response:**

Figure 4: We have changed the representation of the model horizontal grid points from a grid of lines to black crosses as stated in the response, which significantly improves the distinction from the red partition boxes.

**2. Figure 10 Issues:**

**- Visibility: The black line indicating the cross-section in Fig. 10a is obscured by the wind vectors. Please use a contrasting color (e.g., gray or magenta) or increase the line thickness.**

Response:

We have changed the wind vector arrows from black to a high-contrast purple to make the overlaid black cross-section line more visible without altering the line itself, maintaining consistency across subplots.

**- Physical Interpretation: The cross-section passes through the typhoon center. It is physically puzzling why the u-wind analysis increments (Fig. 10b) appear overwhelmingly positive across the center.**

Response:

We apologize for the confusion. Figure 10b plots the increment in total wind speed, not the u-component. The background field underestimated the wind speed in the typhoon core (Fig. 9c), so a positive increment across the center is physically consistent for wind speed magnitude.

**- Mismatches: The caption for Fig. 10d states it shows v-wind, but the unit label in the image is $\pi\pi$ (dimensionless pressure). Additionally, the caption describes panels up to (j), but the layout and labels need to be checked for consistency.**

Response:

We have corrected the caption and verified all labels. Panel (d) correctly shows the $\pi$ (Exner pressure)

increment. The reference to non-existent panels (j) has been removed.

**3. Section Titles: The titles for Section 4.2 and Section 4.3 are identical ("Results of OSSE Experiment"). Section 4.3 discusses real-data assimilation and should be titled accordingly (e.g., "Results of Real-Data Assimilation Experiment").**

Response:

Section Titles: The title of Section 4.3 has been changed to "4.3 Assimilation of Sea Surface Wind Observations".

**4. Figure 11 Caption: The caption is too brief. It needs to be expanded to clearly describe what panels (a) and (b) represent (e.g., "Comparison of typhoon tracks (a) and maximum wind speed (b) between CTL and DA experiments...").**

Response:

Figure 11 Caption: The caption has been expanded as suggested: " Figure 1: Comparison of (a) typhoon tracks and (b) 10-m maximum wind speed (units: m s⁻¹) among the control experiment (CTL), the data assimilation experiments (DA1, DA2, DA3), and observations (OBS) for Typhoon Yagi (2024). Forecasts start at 00:00 UTC on 6 September 2024."

**5. Typos and Grammar:**

**- Line 16:** "This paper constructed..." → "**constructs**" (or presents/proposes).

**- Line 83:** "microphysical schem" → "**scheme**".

**- Line 261:** "Figure 5a showed..." → "**shows**".

**- Line 286:** "The results shows that..." → "The results **show** that...".

**- Line 445:** "assimilate atmospheric variable" → "atmospheric **variables**".

Response:

All noted corrections have been made in the manuscript ("constructs", "scheme", "shows", "show", "variables"). We have also performed a thorough check for tense and grammar throughout the paper.